# Monitoring Knee Contact Force with Force-Sensing Insoles

**DOI:** 10.3390/s23104900

**Published:** 2023-05-19

**Authors:** Alex Spencer, Michael Samaan, Brian Noehren

**Affiliations:** 1Department of Physical Therapy, College of Health Sciences, University of Kentucky, Lexington, KY 40508, USA; 2Department of Kinesiology & Health Promotion, College of Education, University of Kentucky, Lexington, KY 40508, USA; 3Department of Biomedical Engineering, College of Engineering, University of Kentucky, Lexington, KY 40508, USA; 4Department of Orthopedic Surgery and Sports Medicine, College of Medicine, University of Kentucky, Lexington, KY 40508, USA

**Keywords:** wearable, knee contact force, musculoskeletal modeling, statistical modeling, tissue loading

## Abstract

Numerous applications exist for monitoring knee contact force (KCF) throughout activities of daily living. However, the ability to estimate these forces is restricted to a laboratory setting. The purposes of this study are to develop KCF metric estimation models and explore the feasibility of monitoring KCF metrics via surrogate measures derived from force-sensing insole data. Nine healthy subjects (3F, age 27 ± 5 years, mass 74.8 ± 11.8 kg, height 1.7 ± 0.08 m) walked at multiple speeds (0.8–1.6 m/s) on an instrumented treadmill. Thirteen insole force features were calculated as potential predictors of peak KCF and KCF impulse per step, estimated with musculoskeletal modeling. The error was calculated with median symmetric accuracy. Pearson product-moment correlation coefficients defined the relationship between variables. Models develop per-limb demonstrated lower prediction error than those developed per-subject (KCF impulse: 2.2% vs 3.4%; peak KCF: 3.50% vs. 6.5%, respectively). Many insole features are moderately to strongly associated with peak KCF, but not KCF impulse across the group. We present methods to directly estimate and monitor changes in KCF using instrumented insoles. Our results carry promising implications for internal tissue loads monitoring outside of a laboratory with wearable sensors.

## 1. Introduction

Applications for internal tissue load measurement include the treatment evaluation of orthopedic injuries, management or avoidance of overuse injuries, and study of how chronic loading influences tissue health. Currently, there are no methods to monitor these loads in pragmatic settings. While instrumented prostheses or other implantable devices provide the gold standard for accurately measuring internal loads [1,2,3,4,5,6,7], their invasive nature limits their ability to be widely used for clinical problem understanding. Alternatively, although noninvasive laboratory-based methods such as the estimation of intersegmental moments and forces with inverse dynamics [8,9,10,11,12], joint contact forces with musculoskeletal modeling [13,14,15,16,17,18,19,20,21,22,23,24,25,26,27,28,29,30,31,32], or tissue stress and strain with finite element analysis [13,14,15,16,17,18,19,20,21,22,23] provide substantial analytical depth, the laboratory-based instrumentation required limits their applicability in a clinic or in patients’ activities of daily living. The liberation of internal tissue load monitoring outside of a traditional laboratory would allow rehabilitation specialists to optimize patient treatment with more precision in real-world settings. 

One potential avenue to estimate internal tissue loads in real-world scenarios is with wearable sensors. In particular, force-sensing insoles that estimate the normal component of foot–shoe contact force have received attention due to their ease of use, unobtrusiveness, and potential to help answer important research questions [24,25]. Through the application of statistical methods, internal tissue loading in the lower body could be monitored by measuring only this foot contact force. However, there are some important considerations to be made when developing models for clinical populations. First, the intended population will impact the potential generalizability of the model across people or even between limbs. For example, post-surgical patients that demonstrate significant limb asymmetry may require model calibration per limb, while a healthy population with sufficient between-limb symmetry may not require specific calibration. Similarly, post-surgical populations that demonstrate higher between-person kinematic and kinetic variability during gait may require calibration per subject. Second, the intent of these models could be to either directly estimate or monitor tissue loading change over time with surrogate measures. Direct estimation would potentially provide a more precise solution to internal load monitoring, but surrogate measures would allow researchers and clinicians to monitor changes over time (e.g., throughout a treatment plan) while providing potentially increased model generalizability across a population. In this study, we explore the potential for all permutations of the considerations listed above (model per-subject versus per-limb and direct estimation versus surrogate measures).

The first purpose of this study is to develop models with varying levels of specificity (per-limb and per-subject) that directly estimate peak KCF and KCF impulse per step with data from force-sensing insoles across a range of speeds from a healthy population. We hypothesize that models developed per-limb will perform similarly as those developed per-subject for this healthy cohort. The second purpose is to explore the potential to monitor KCF metrics (peak and impulse) with surrogate measures (foot contact force data features). 

## 2. Materials and Methods

This study was approved by the University of Kentucky Institutional Review Board. Nine subjects (3F/6M, age 27 ± 5 years, mass 74.8 ± 11.8 kg, height 1.74 ± 0.08 m) provided their written informed consent and were enrolled in the study. Only subjects meeting the following inclusion criteria were considered for the study: Tegner score equal to or greater than 4; 15–40 years of age; Body Mass Index 18–25 kg/m^2^; participate in competitive sport or run at least 10 miles per week; and no history of movement impairment or lower extremity injury. Finally, subjects were excluded if they had a history of previous surgeries. We chose nine participants for this study for the purpose of establishing feasibility in the proposed approaches. 

The motion capture protocol was consistent with previously published methods [26]. Fifty-two retroreflective markers were placed on each subject (25 as tracking clusters and 27 on anatomical landmarks). Marker locations were collected at 200 Hz with a 12-camera motion capture system (Motion Analysis, Santa Rosa, CA, USA) simultaneously with force plate data at 1200 Hz from a dual-belt instrumented treadmill (Bertec Corporation, Columbus, OH) as subjects walked at five different speeds (0.8, 1.0, 1.2, 1.4, and 1.6 m/s) for 60 s each. Marker position and force plate data were filtered with 4th order low-pass Butterworth filters at 8 and 35 Hz, respectively. Foot contact force data was collected from each condition using single sensor loadsol^®^ insoles (Novel Electronics, St. Paul, MN, USA) at 100 Hz. All subjects wore New Balance WR662 running shoes (New Balance, Brighton, MA, USA).

In order to sync the motion capture and foot contact force data, each trial began with a right-foot stomp while the treadmill was stopped followed by a controlled increase in speed until the condition speed was achieved. Force plate and insole force peaks corresponding to the stomp were semi-automatically identified and matched. The data was then scanned across a +/− 50 ms window of these peaks to optimize the synchronization by maximizing the cross-correlation of the foot contact force (force-sensing insole) and ground reaction force (force plate) data. Stance intervals were defined with 20 N thresholds from the insole data. 

Knee contact forces were estimated using the Gait2392 model in OpenSim (version 4.0) [27]. The Gait2392 musculoskeletal model consists of 23 degrees of freedom (DOF) and 92 muscles, with the knee restricted as a 1 DOF hinge joint (flexion/extension). Model weight, height, and segment lengths were scaled per subject from static trial marker positions. Muscle forces were estimated via static optimization [28], then used within the joint reaction analysis tool to estimate KCF expressed in the tibial reference frame [29].The peak KCF and KCF impulse were computed as the maximum and time integral of the resultant KCF vector magnitude per stance phase, respectively. 

A total of 13 foot contact force features were extracted per step from the insoles [Table 1]. Traditional features such as stance time, peak force, and loading rate were supplemented with non-traditional metrics that may generate deeper insight into subtle differences in gait mechanics which are not readily captured by individual metrics. Equations, illustrations, and descriptions for each of these features are provided in Appendix A. 

Knee contact force prediction models were created per-limb and per-subject using stance phases across all walking speeds performed. A total of 6634 steps were collected across subjects, with an average of 185 steps per limb. No steps were excluded from analysis. All features were first z-score normalized to ensure a zero-mean and unit variance, then used as inputs to a linear regression model that was trained with a 10-fold cross validation scheme stratified by walking speed. The median symmetric accuracy (MSA) was chosen as the evaluation metric, as it has been shown to produce unbiased and robust models while maintaining a translatable output (percent error) (Equation (1)) [30]. Individual predictors for the final models were chosen through a best subset selection method. The overall method of KCF prediction is illustrated in Figure 1.
(1)$$\mathrm{MSA}=100\left(e^{\left(M\left(\left|\ln\;\left(Q\right)\;\right|\right)\right)}-1\right)$$
where $Q=\frac{\mathrm{prediction}}{\mathrm{observation}}$ and M is the median operator.

Pearson product–moment correlation coefficients define the relationship between foot contact force features and KCF metrics. The following ranges define the strength of correlations: strong—r ≥ 0.7; moderate—0.5 ≤ r < 0.7; weak—0.3 ≤ r < 0.5; and negligible—r < 0.3. These correlation coefficients were calculated per subject per limb, then averaged across the group.

## 3. Results

The KCF prediction error is lower in models developed per-limb (2.19% for KCF Impulse and 3.50% for peak KCF) than those developed per-subject (3.40% for KCF Impulse and 6.47% for peak KCF) [Table 2]. The error is consistently lower for the KCF Impulse than peak KCF. Finally, the models utilize a relatively low number of features, indicating good computational feasibility for these methods [Table 2]. 

The correlation analysis identifies a number of insole features which are moderately to strongly associated with peak KCF (7 strong, 4 moderate, and 2 negligible) [Table 3]. All insole features are negligibly correlated with KCF impulse at a group level. However, a number of correlations re moderate to strong on a per-limb basis, as illustrated in Appendix B. 

## 4. Discussion

This study has two primary purposes: (1) develop models of varying specificity to directly estimate peak KCF and KCF impulse per step; and (2) establish the feasibility of using foot contact force data features as surrogate measures for KCF metrics. We find that models developed per-limb produce a lower error than those developed per-subject. Additionally, we identify a number of insole features which re moderately to strongly associated with peak KCF (7 strong, 4 moderate, and 2 negligible). However, while all insole features are negligibly correlated with KCF impulse at a group level, there re individual strong and moderate correlations on a per-limb basis. These results demonstrate the feasibility of monitoring KCFs outside of a traditional laboratory with wearable sensors. 

The prediction error from the direct estimation models developed in this study demonstrates that a single wearable sensor can produce accurate KCF estimates. These models typically utilize between five and nine predictors [Table 2], which demonstrates a computational efficiency balanced with robust biomechanical representation from the features engineered. The collection of insole force features [Appendix A] is designed to capture subtle gait mechanics which are not readily captured by individual metrics. For example, traditional ground reaction force metrics measured through different phases of stance (weight acceptance and propulsive) have been previously found to correlate with walking kinematics such as knee flexion excursion [31]. Additionally, mean pseudo-frequency has been shown to discriminate between rearfoot and non-rearfoot patterns [32]. Combining features that capture these movement characteristics results in an accurate KCF estimation through robust biomechanical representation. 

The moderate to strong correlations identified suggest that individual insole force features can be used as surrogate measures to monitor peak KCF. For example, given the consistently strong relationship between insole peak force and peak KCF, this metric could be used to monitor peak KCF changes through time, although it is not a direct estimate. Alternatively, the relationships between KCF impulse and all insole features re negligible at a group level. The group-wide negligible relationships stem from the variability in the relationship directions between subjects [See Appendix B]. For example, the average correlation between the KCF impulse and insole loading rate is 0.11, but with a range of −0.75 to 0.71. We speculate that the variability in correlation direction stems from differences in how subjects’ gait changes with speed. As illustrated in Figure 2, Subject A develops a significant peak in their KCF curve during the first half of stance as the speed increases, while Subject B does not. These individual gait differences require relationships between force-sensing insole features and KCF impulse to be calibrated per-limb or per-subject. Future research and clinical use of these methods could perform a set of walking trials to establish relationships between insole force features and KCF impulse for each patient. Conversely, the results from this study suggest that these trials would not be necessary to monitor peak KCF with select insole force-based surrogate features.

While the models from this study use data from a healthy population, the methods could be extended to patients following surgery or with movement pathology. For example, clinicians could mitigate the risk of premature osteoarthritis development in anterior cruciate ligament reconstruction patients through the restoration and monitoring of knee loads during activities of daily living. Additionally, gait retraining and subsequent load monitoring for total knee arthroplasty patients could optimize the long-term health of their own tissues and the artificial joint. One future consideration is that while the models developed per-limb from this healthy group only perform moderately better than per-subject models, the error of models developed per-subject would likely increase as the methods extend to post-surgical patients with greater between-limb KCF differences. 

There are a number of limitations to this study. First, because the external validity of these types of models are limited to the training data from which they are developed, the scope of KCF prediction from this study is limited to flat-surface, straight-line walking. Future studies could implement various inclines, steps, and turns on multiple surfaces to increase model generalizability. Additionally, these models are developed from a single data collection. Collecting data on multiple days would provide further generalizability through the introduction of controlled variability in the training data. Third, we chose to utilize a simple linear regression model for this feasibility study. There are several more sophisticated modelling strategies that could be implemented to improve prediction accuracy which have shown strong performance in other fields as demonstrated in [33,34,35,36,37]. Finally, although EMG-informed models have been shown to be the most effective option to estimate KCFs with musculoskeletal modeling [38,39,40], the KCF results in this study are consistent with those previously reported [28,39] and are generated with a validated musculoskeletal model and muscle force estimation method. 

## 5. Conclusions

We present methods which can be used to monitor KCF metrics during activities of daily living using force-sensing insoles. While these sensors have been previously used to monitor rehabilitation progress in various ways, no studies to date have used them to estimate musculoskeletal-model generated KCF during walking. The performance of both the per-limb and per-subject based models developed indicates that accurately estimating KCF metrics can be done with a single wearable sensor. Further, we identify a number of insole features which are strongly or moderately associated with peak KCF, but not KCF impulse. These results carry promising implications for the estimation of KCFs outside of a traditional laboratory with wearable sensors. 

## Figures and Tables

**Figure 1 sensors-23-04900-f001:**
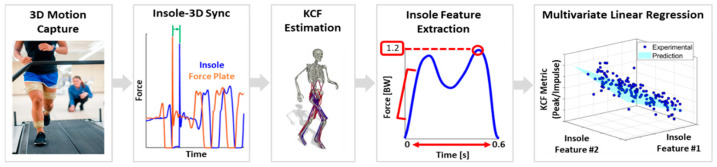
Overall method of KCF prediction with force-sensing insole data. Dual force traces in Insole-3D Sync represent data from left & right foot insoles.

**Figure 2 sensors-23-04900-f002:**
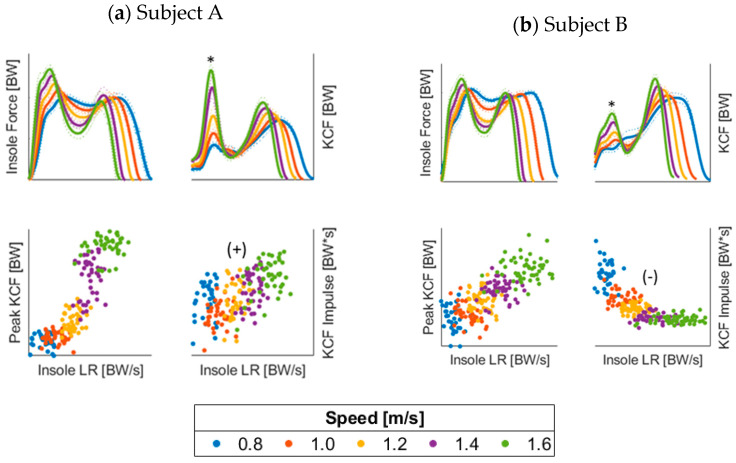
Comparison of two subjects’ KCF evolution as speed increases. Subject A develops a much larger first peak (*) in KCF than Subject B as speed increases, possibly driving the opposite direction of the association between insole and KCF metrics. LR = loading rate.

**Table 1 sensors-23-04900-t001:** Features extracted from insole data per step.

Time Domain—Traditional	Time Domain—Other	(Pseudo) Frequency Domain
Stance Time [s]	Skewness	DFT Max [Hz]
Peak Magnitude [BW]	Mid-Drop Peak Magnitude	Mean PF
Impulse [BW×s]	WAC Impulse [BW×s]	
Loading Rate [BW/s]	Prop. Impulse [BW×s]	
IP Magnitude [BW]	WAC/Prop. Impulse Symmetry	

**Note:** IP = impact peak; WAC = weight acceptance; Prop. = Propulsive; DFT = Discrete Fourier Transform; PF = Pseudo-Frequency; and BW = bodyweight.

**Table 2 sensors-23-04900-t002:** Performance of KCF prediction models.

	KCF Impulse		Peak KCF	
	MSA [%]	Num. Predictors	MSA [%]	# Predictors
Per Limb	2.19 (1.65–2.54)	8.0 (7.0–8.8)	3.50 (2.78–5.09)	7.0 (5.3–8.0)
Per Subject	3.40 (2.87–4.24)	9.0 (7.0–9.0)	6.47 (5.07–11.06)	8.0 (7.0–9.0)

**Note:** MSA = median symmetric accuracy. All values are reported as median (interquartile range).

**Table 3 sensors-23-04900-t003:** Correlation coefficients between insole features and KCF metrics.

		Peak KCF	KCF Impulse
Time Domain [Traditional]	Stance Time [s]	−0.84 ± 0.05 ***	0.09 ± 0.48
	Peak [BW]	0.87 ± 0.09 ***	0.14 ± 0.40
	Impulse [BW×s]	−0.76 ± 0.07 ***	0.11 ± 0.48
	Loading Rate [BW/s]	0.80 ± 0.24 ***	0.11 ± 0.44
	IP Magnitude [BW]	0.62 ± 0.16 **	0.09 ± 0.34
Time Domain [Other]	Skewness	0.81 ± 0.12 ***	0.20 ± 0.41
	Mid-Drop Peak	−0.82 ± 0.12 ***	−0.10 ± 0.46
	WAC Imp [BW×s]	−0.05 ± 0.34	0.08 ± 0.19
	Prop Imp [BW×s]	−0.66 ± 0.14 **	0.09 ± 0.42
	WAC/Prop Impulse	0.56 ± 0.17 **	−0.02 ± 0.33
	Symmetry	−0.24 ± 0.44	−0.03 ± 0.25
(Pseudo) Frequency Domain	DFT Max [Hz]	0.63 ± 0.09 **	−0.14 ± 0.42
	Mean PF	0.73 ± 0.12 ***	−0.14 ± 0.45

**Note**: Values are presented as mean ± standard deviation of all speed conditions per subject per limb. *** = strong correlation (|r| ≥ 0.7). ** = moderate correlation (0.5 ≤ |r| < 0.7).

## Data Availability

The data presented in this study are available on request from the corresponding author.

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
