# Peer review of "Monitoring Knee Contact Force with Force-Sensing Insoles"

_sensors, 2023, doi:10.3390/s23104900_

Round 1
Reviewer 1 Report
The paper presents the feasibility study for monitoring of knee contact force, indirectly, based on force-sensing insole data. The limited group of nine healthy volunteers were examined during the walking at five different speeds on treadmill.
Minor remarks:
Although references [28] and [29] are cited, more details about the calculation of peak KCF and impulse KCF is recommended to make the methodological procedure completely clear.
Features extracted from insole data per step. How many steps were available? Are some steps excluded from the analysis?
Citing references in the Conclusion section is unusual (references [25], [37], [38]). This should be moved to the Discussion section.
Appendix contains the following paragraph that should be removed: “The appendix is an optional section that can contain details and data supplemental to the main text—for example, explanations of experimental details that would disrupt the flow of the main text but nonetheless remain crucial to understanding and reproducing the research shown; figures of replicates for experiments of which representative data is shown in the main text can be added here if brief, or as Supplementary data. Mathematical proofs of results not central to the paper can be added as an appendix.”
Appendix B could be divided to three sections and each section could have a) and b) part: Section I with title “KFC and Traditional Metrics” and two subsections “a) Peak KFC and Traditional Metrics” and “b) KFC Impulse and Traditional Metrics”, Section II with title “KFC and Non-Traditional Metrics” including a) and b) subsections similarly as section I, and Section III with title “KFC and (Pseudo)-Frequency Domain Features” including a) and b) subsections.
Reviewer 2 Report
I cannot give any specific recommendations for the paper improvement as I work outside of its direct area.
However I can attest that the design of experiments, used instrumentation, experimental results all look credible and appropriately described. The paper is presented in a logical and well structured manner, with useful appendices and extensive references.
I ticked the box for fine English as I could not spot any mistakes (however I am not a native speaker).
Reviewer 3 Report
The manuscript is centred on a very interesting and timely topic, which is also quite relevant to the themes of Sensors
The authors use insole sensors for modelling estimations of metrics of knee contact force (KFC): peak and impulse. The estimations are based on the musculoskeletal model Gait2392. They use coefficient correlations between KFC peak and the impulse to establish insole features associated with these KFC metrics. The interest of the work is the method allows monitoring the patients outside of laboratories.
The authors do an exhaustive analysis of the registered data in the experiments. The experiments are perfectly exposed, and the results and conclusions are clear. The structure is straightforward. The objectives and methodology are well described. These are some comments that I suggest should be considered to clarify some issues:
The authors argue the interest in reviewing current works on this topic. The structure is straightforward. The objectives and methodology are well described. These are some comments that I suggest should be considered to clarify some issues:
-
References to works in the last 5 years are not included, I suggest including references to works based on statistical techniques in the last 5 years, as opposed to machine learning techniques used in related problems.
-
The study is carried out on 9 individuals. I suggest that it be explained why the sample size chosen is adequate and if it offers guarantees that the results can be generalized.
-
Review the title of the first column of table 1 and the equation of table 2.
